# Features of "false positive" unruptured intracranial aneurysms on screening magnetic resonance angiography

Minsu Jang[1], Jang Hun Kim[1,2]*, Jin Woo Park[1], Haewon Roh[3], Han-Joo Lee[1], Junghan Seo[1], Sung Hwan Hwang[1], Joon Ho Yoon[1], Sang Hoon Yoon[1], Byung-Kyu Cho[1]

1 Department of Neurosurgery, Armed Forces Capital Hospital, Seongnam-si, Gyeonggi-do, Republic of Korea, 2 Trauma Center, Armed Forces Capital Hospital, Seongnam-si, Gyeonggi-do, Republic of Korea, 3 Department of Neurosurgery, Guro Hospital, Korea University College of Medicine, Seoul, Republic of Korea

* jhkimns@naver.com

## Abstract

### Background

Physicians can find it challenging to decide whether confirmative digital subtraction angiography (DSA) should be performed in patients who present with "suspicious small aneurysm-like structures" on magnetic resonance angiography (MRA). Factors associated with "false positive aneurysms on MRA" (FPAMs)," which are finally confirmed as negative on DSA, have rarely been reported. This study aimed to identify the clinical or radiologic clues indicative of FPAM on DSA.

### Methods

Patients who had undergone DSA between 2016 and 2019 for suspicious aneurysm-like structures < 5 mm in size on MRA were enrolled. Patient demographics and the details regarding the geometry of the structures were retrospectively reviewed. Univariate and multivariate logistic regression analyses were conducted to identify the associated factors. Receiver operating characteristic curve analysis was performed to assess the clinical implications.

### Results

Of the 107 suspicious structures, 46 were indicated as being false positive on DSA (42.96%). Location (positive on C7 and negative on C5-6 ICA) and lower dome to neck ratio were found to be significant parameters in the multivariate analysis. The dome to neck ratio threshold value was 0.99.

### Conclusion

Suspicious aneurysm-like structures located not on C5-6 but on C7 ICA and having wide neck morphologies (dome to neck ratio < 0.99) are highly likely to be negative on DSA.

**Data Availability Statement:** All relevant data are within the manuscript and its Supporting Information files.

**Funding:** This work was supported by the Korean Military Medical Research Project funded by the ROK Ministry of National Defense.

**Competing interests:** The authors have declared that no competing interests exist.

## Introduction

Noninvasive magnetic resonance angiography (MRA) is highly sensitive for detecting unruptured intracranial aneurysms (UIAs) and is therefore commonly used as an initial screening tool for investigating the cerebral vasculature. [1] While digital subtraction angiography (DSA) is considered to be the gold standard for evaluating UIAs, it is usually undertaken to confirm the final diagnosis due to its invasiveness, cost, and hospitalization-related inconveniences to patients. [2] As screening MRA is being widely used, clinicians frequently encounter suspicious aneurysm-like structures, especially when these are too small or have a wide neck morphology. In such cases, subsequent DSA is necessary to determine whether these structures are true aneurysms. These suspicious structures could finally be diagnosed as either "true positives" or "false positive aneurysms on MRA" (FPAMs) depending on the DSA results. In a recent meta-analysis, it was found that the FPAM rate was 82% when aneurysms were <3 mm in size. [3]

It can be challenging for physicians to decide whether to perform DSA when patients present with suspicious aneurysm-like structures on screening angiographies. In the past, these small suspicious structures were assessed on regular outpatient follow-up without performing DSA. [4] However, small aneurysms, for which treatment was not recommended previously, are widely accepted as treatment indications according to recent evidence. [5, 6] For military service personnel, a confirmatory diagnosis of UIAs is routinely performed, because the physical capability of these individuals needs to be graded definitively. Therefore, even as the number of individuals undergoing DSA is increasing, efforts must be taken to reduce unnecessary DSA procedures for these small suspicious aneurysm-like structures.

To date, studies that have examined the relationship between MRA and DSA findings have focused on the sensitivity or false-positive rate of screening tools. [7–9] Herein, we designed a retrospective study to determine clinical or radiological clues that can identify suspicious aneurysm-like structures as FPAMs. Clinicians could then use these preexisting factors to decide whether DSA is necessary.

## Methods

### Patients selection and data acquisition

This study was designed as a retrospective analysis and was approved by the Institutional Review Board of the Human Research Center in Armed Forces Capital Hospital (AFCA-19-IRB-030). The requirement for informed consent was waived due to the retrospective nature of the study.

Inclusion criteria were as follows:

1. patients who had a saccular UIA <5 mm in maximal size as observed on MRA,

2. patients who underwent DSA for a definitive diagnosis between August 2016 and July 2019.

Ruptured aneurysms and aneurysms related to Moyamoya disease, brain arteriovenous malformations, or fusiform dilation of shape were excluded. In total, 107 aneurysms in 92 patients were included. A flow chart of the enrollment process is shown in Fig 1. General information regarding patient age, sex, time period between MRA and DSA, and other relevant history was retrospectively collected.

### Radiologic evaluations

Every patient underwent time-of-flight (TOF) sequence 3-T MRA (DISCOVERY MR750; GE Healthcare, Chicago, IL, USA) for screening and DSA (Allura clarity FD2015, Phillips, Best,

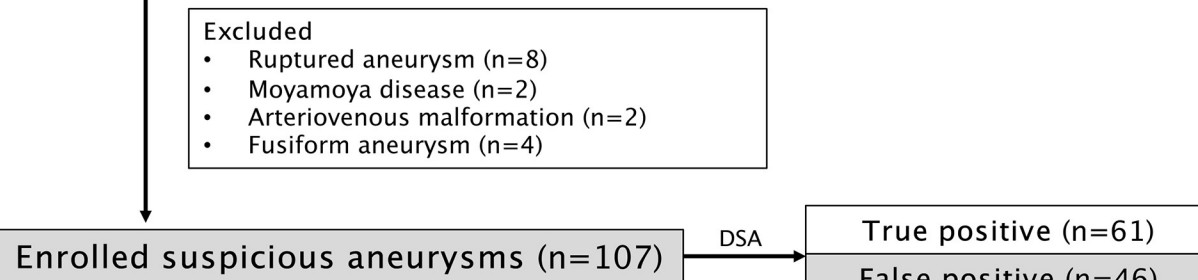

Reviewed "Suspicious aneurysm–like structures" (n=123)
- Patients who had a saccular UIA <5 mm in maximal size as observed on MRA,
- and underwent DSA for a definitive diagnosis between August 2016 and July 2019

Excluded
- Ruptured aneurysm (n=8)
- Moyamoya disease (n=2)
- Arteriovenous malformation (n=2)
- Fusiform aneurysm (n=4)

Enrolled suspicious aneurysms (n=107) — DSA →

True positive (n=61)

False positive (n=46)

**Fig 1. Flow chart of the enrollment and classification process.**

Netherlands) for confirmative evaluation. Suspicious aneurysm-like structures were identified by a neuroradiologist and a vascular neurosurgeon before DSA was performed. Any disagreements were mediated by a senior neurosurgeon.

The images were retrospectively reviewed using the Maroview picture archiving communication system (Marotech Inc., Seoul, Republic of Korea). In particular, the detailed geometry of the aneurysm-like structure, in terms of its height, width, neck size, maximal diameter, aspect ratio, and dome to neck ratio, was assessed based on the imaging data (Fig 2). To

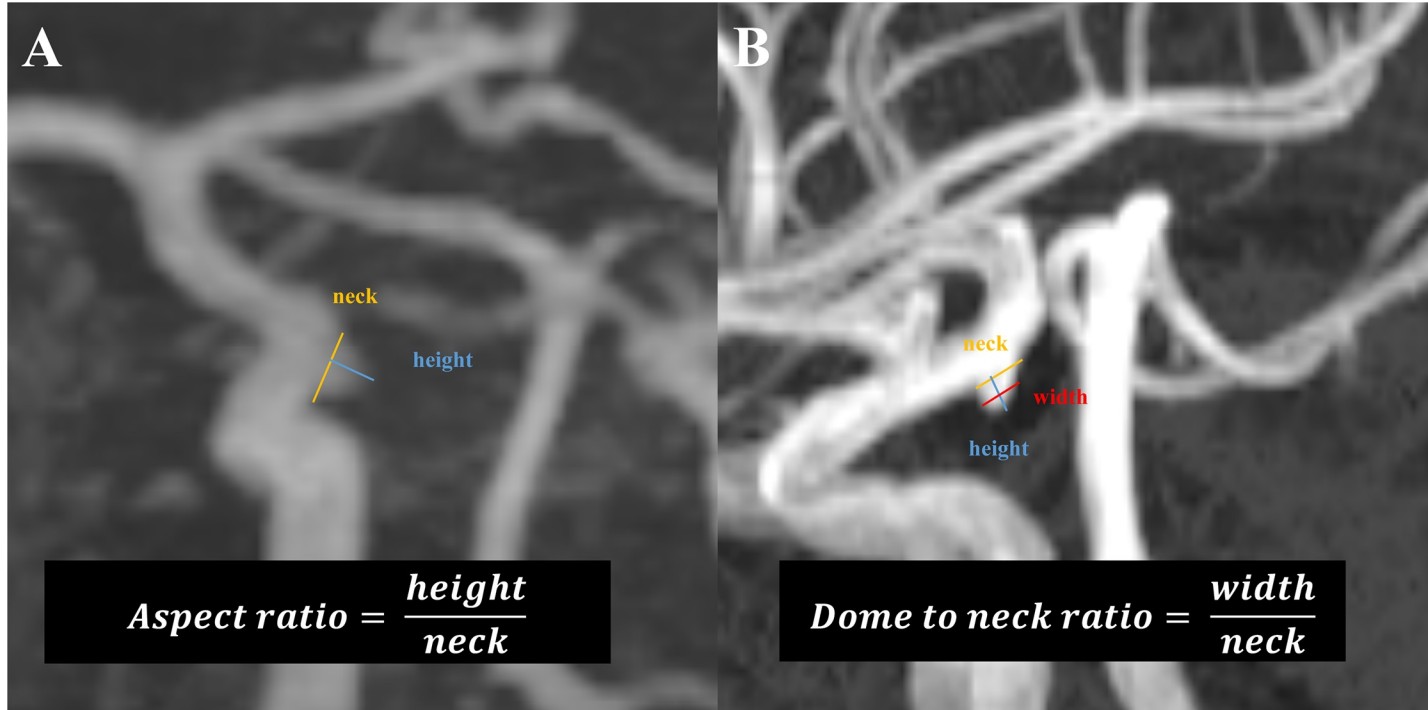

**Fig 2. Measurement of the geometrical parameters.** Aneurysm-like structures were selected and then the image was accordingly magnified. The height, neck, width, and maximal diameter were directly measured. The aspect ratio (A) and dome to neck ratio (B) were calculated using standard formulas. Dome to neck ratio is measured from the center of the neck to the top of the aneurysm dome, and the aneurysm width is measured perpendicular to the dome to neck line.

minimize errors, every parameter was independently measured twice by two individuals (JH Kim & MS Jang), and the mean value was used. An aneurysm-like structure was definitively judged as a 'true positive aneurysm' or 'FPAM' according to the 3-dimensional reconstructed image based on rotational angiography results (Fig 3). Judgements regarding the radiologic images were separately made by two physicians (JH Kim & MS Jang) who were blinded to each other's determinations. No discrepancies were noted.

### Data analysis

Continuous data are expressed as means ± standard deviations, and categorical data are reported as frequencies and percentages. Univariate and multivariate logistic regression analyses were conducted to identify factors associated with FPAM using SPSS (version 23.0, IBM, Chicago, IL, USA). A $p$-value of $< 0.05$ was considered statistically significant. To ascertain the threshold values regarding the geometry of the aneurysm-like structures, receiver operating characteristic (ROC) curve analysis was performed using the continuous variables.

## Results

In total, 107 aneurysm-like structures in 92 patients were included in the analyses. The baseline characteristics of patients and suspicious structures are presented in Table 1. Due to the study being conducted in a military hospital, most of the patients were healthy, young, male soldiers. The majority of the aneurysm-like structures were small, with a mean maximal diameter and height of 2.96 mm and 2.15 mm, respectively.

The results of the univariate and multivariate logistic regression analyses are shown in Tables 2 and 3. Of the 107 aneurysms, 46 were diagnosed as FPAMs using DSA (false-positive rate: 43.0%). In the univariate analysis, anatomical location and geometric factors, including smaller maximal diameter, height, width, and aspect and dome to neck ratios, were found to be associated with FPAMs. In the multivariate analysis (Table 3), location on the C7 ICA and not on the C5-6 ICA ($p<0.001$) and a smaller dome to neck ratio ($p<0.001$) were found to be significantly associated with FPAMs.

Table 4 and Fig 4 display the ROC curve analysis results regarding the dome to neck ratio, for which the threshold value was 0.99.

## Discussion

The majority of FPAMs were located on the C7 ICA (29/46, 63.0%) and had a low dome to neck ratio (0.77±0.15); these were identified as significant associated factors in multivariate analysis. Thus, suspicious aneurysm-like structures located on the C7 ICA, not on C5-6 ICA, and with dome to neck ratios less than 0.99 are highly likely to be false positive on DSA.

### Discrepancy between MRA and DSA

CTA and MRA have widely been used as the primary diagnostic tools for detecting UIAs. [10, 11] They are both sensitive and have an acceptable level of specificity. [3, 10] Usually, CTA is used to diagnose ruptured cerebral aneurysms with high sensitivity, while MRA is used to screen unruptured aneurysms in patients for whom radiation exposure needs to be avoided. [12] Radiologists and neurosurgeons can find it challenging to decide on a management plan when they encounter small suspicious aneurysms on the ICA terminus. It can be challenging to determine whether these aneurysm-like structures on MRA are junctional dilations (infundibula, Fig 5) or true positive aneurysms. [13]

Some signs suggestive of an aneurysm are as follows:

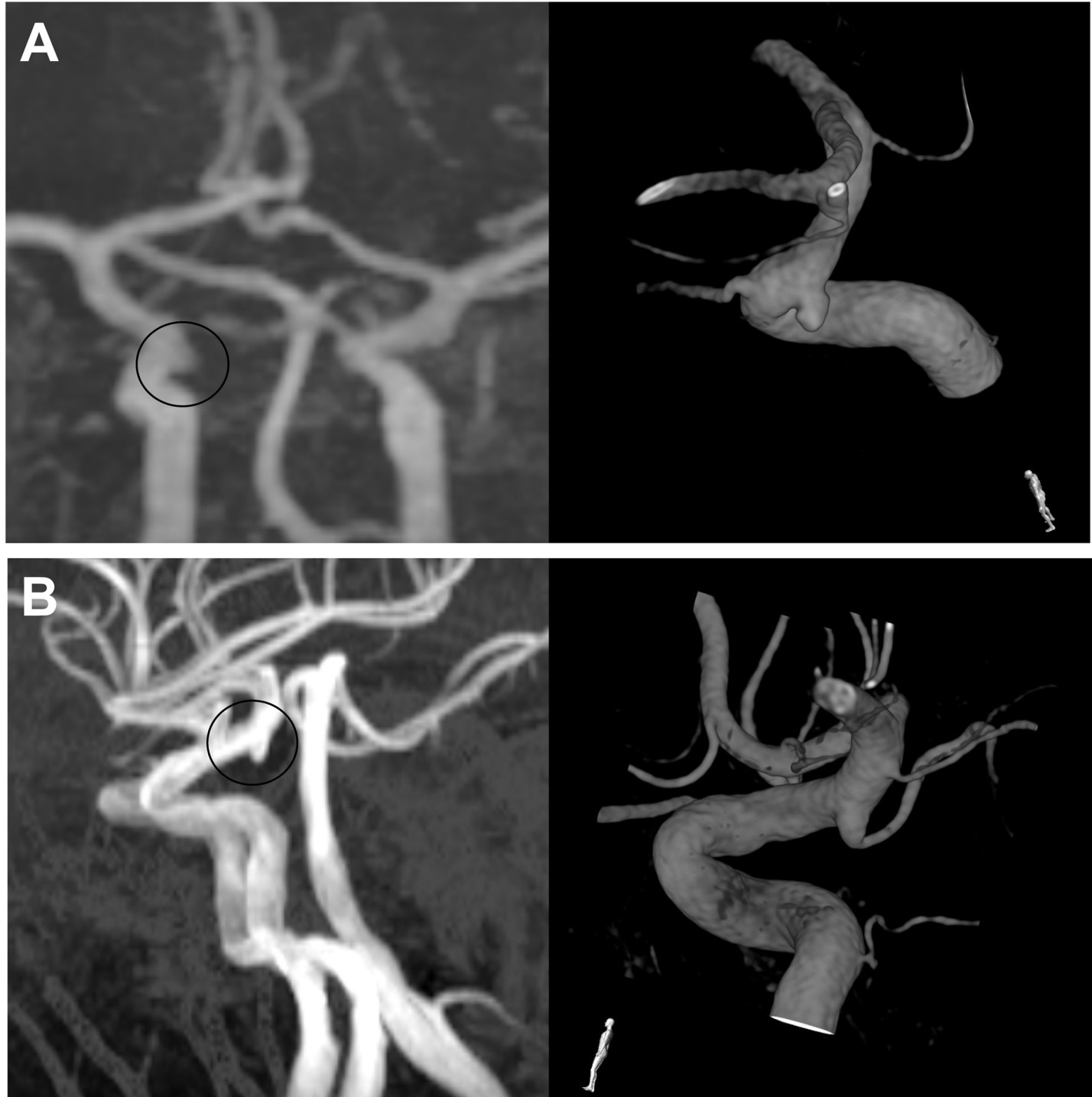

**Fig 3.** Illustrative images of true positive (A) and false positive aneurysms (B) on DSA. Both MRA images (left) reveal suspicious aneurysm-like structures on the ICA terminus. However, catheterized angiography results are contradictory. A small side-wall aneurysm is observed in (A), while the infundibulum of the PCoA is shown in (B). MRA, magnetic resonance angiography; ICA, internal carotid artery; PCoA, posterior communicating artery.

**Table 1. Baseline characteristics of the included patients and aneurysms.**

| Patients (n = 92) | |
|---|---|
| Females | 4 (4.35%) |
| Age (years) | 36.3±13.63 |
| MRA to DSA (months) | 3.0±2.67 |
| History | |
| Hypertension | 9 (9.78%) |
| Diabetes | 1 (1.09%) |
| Smoking | 48 (52.17%) |
| **Aneurysm (n = 107)** | |
| Left side | 66 (61.68%) |
| Maximal diameter (mm) | 2.96±1.05 |
| Height (mm) | 2.15±1.06 |
| Width (mm) | 2.40±1.13 |
| Neck (mm) | 2.60±0.70 |
| Aspect ratio | 0.86±0.46 |
| Dome to neck ratio | 0.94±0.44 |

MRA, magnetic resonance angiography; DSA, digital subtraction angiography

**Table 2. Demographics and results of the univariate logistic regression analysis.**

| Parameters | False positive (n = 46) | True positive (n = 61) | Univariate analysis | | |
|---|---|---|---|---|---|
| | | | Hazard ratio | 95% Confidence interval | *p*-value |
| **Side** | | | 0.942 | 0.429–2.067 | 0.881 |
| Left | 28 (60.9%) | 38 (62.3%) | | | |
| Right | 18 (39.1%) | 23 (37.7%) | | | |
| **Location [a]** | | | | | |
| Clinoid and ophthalmic ICA (C5-6) | 4 (8.2%) | 38 (62.3%) | 1 | | **<0.001**[*] |
| Communicating ICA (C7) | 29 (63.0%) | 7 (11.5%) | 0.025 | 0.07–0.095 | **<0.001**[*] |
| Others | 13 (28.3%) | 16 (26.2%) | 0.130 | 0.037–0.458 | **0.002**[*] |
| Cavernous ICA (C4) | 1 (2.2%) | 3 (4.9%) | | | |
| AChA | 4 (8.7%) | 4 (6.6%) | | | |
| MCAB/ICAB | 2 (4.3%) | 5 (8.2%) | | | |
| Posterior circulation | 3 (6.5%) | 4 (6.6%) | | | |
| Distal ACA | 3 (6.5%) | 0 (0%) | | | |
| **Type** | | | | | |
| Side-wall | 2 (4.3%) | 10 (16.4%) | 1.00 | | 0.158 |
| Major artery | 35 (76.1%) | 43 (70.5%) | 0.246 | 0.050–1.196 | 0.082 |
| Bifurcation | 9 (19.6%) | 8 (13.1%) | 0.178 | 0.030–1.067 | 0.059 |
| **Geometry** | | | | | |
| Maximal diameter | 2.57±0.51 | 3.25±1.24 | 2.424 | 1.384–4.244 | **0.002**[*] |
| Height | 1.73±0.49 | 2.47±1.25 | 2.865 | 1.546–5.309 | **0.001**[*] |
| Width | 1.90±0.54 | 2.78±1.31 | 3.519 | 1.775–6.973 | **<0.001**[*] |
| Neck | 2.50±0.56 | 2.68±0.79 | 1.455 | 0.827–2.559 | 0.193 |
| Aspect ratio | 0.72±0.26 | 0.97±0.54 | 5.901 | 1.574–22.124 | **0.008**[*] |
| Dome to neck ratio | 0.77±0.15 | 1.08±0.53 | 79.917 | 7.906–807.797 | **<0.001**[*] |

ICA, internal carotid artery; AChA, anterior communicating artery; MCAB, middle cerebral artery bifurcation; ICAB, internal carotid artery bifurcation; ACA, anterior cerebral artery

[a] ICA location was classified as Bouthillier classification.

[*]*p* <0.05

**Table 3. Results of the multivariate logistic regression analysis of the factors associated with false negative aneurysms.**

| Parameters | Multivariate logistic analysis | | |
|---|---|---|---|
| | Hazard ratio | 95% Confidential index | p-value |
| **Location** | | | |
| Clinoid and ophthalmic ICA (C5-6) | 1 | | **<0.001**\* |
| Communicating ICA (C7) | 23.863 | 5.080~112.086 | **<0.001**\* |
| Others | 0.340 | 0.083~1.393 | 0.134 |
| **Geometry** | | | |
| Maximal diameter | 5.033 | 0.400~63.327 | 0.211 |
| Height | 1.880 | 0.779~4.539 | 0.160 |
| Width | .084 | 0.004~1.960 | 0.123 |
| Aspect ratio | .367 | 0.018~7.533 | 0.516 |
| Dome to neck ratio | 466.309 | 19.003~11442.374 | **<0.001**\* |

\*$p < 0.05$

1. when the aneurysm-like structure is located at normal branching sites, such as the anterior choroidal artery (AChA) or posterior communicating artery (PCoA), the existence of a distinguishable normal branch separated from the aneurysm-like structure may be suggestive of a true positive aneurysm on DSA (Fig 6A)

2. when the aneurysm-like structure appears to have side-wall morphology, such as an aneurysm located on the superior hypophyseal artery, the axial raw data should be carefully interpreted to detect small distal vessels, which negatively indicate an infundibulum (Fig 6B).

However, MRA occasionally fails to detect very narrow normal branching flow. Furthermore, occasionally, the infundibulum of the normal branch presents as a saccular morphologic side-wall aneurysm on the axial raw image as well as the image reconstructed based on MRA data (Fig 6C). Such experiences led us to conduct this study.

TOF-angiography is an MRI technique that can visualize flow within vessels without the necessity of contrast administration. [3] It is based on the phenomenon of flow-related enhancement of spins entering into an image slice. As a result of being unsaturated, these spins present as a bright signal that surrounds stationary spins. An inherent limitation of this technique is that slow flow or flow from a vessel parallel to the scan plane may become desaturated, as with stationary tissue, resulting in signal loss from the vessel. Furthermore, turbulent flow may undergo spin-dephasing and unexpectedly short T2 relaxation, resulting in a long acquisition time. In the case of aneurysms associated with slow, turbulent flow or flow parallel to the spin plane (similar to that in the C7 ICA), discrepancies between imaging findings and

**Table 4. Results of the receiver operating characteristic curve analysis for the dome to neck ratio of the aneurysms.**

| Parameters | Dome to neck ratio |
|---|---|
| Area under the ROC curve | 0.733 |
| Significance level ($p = 0.5$) | <0.001 |
| Youden index | 0.405 |
| Sensitivity | 49.2% |
| Specificity | 91.3% |
| Associated criterion | **0.9887** |

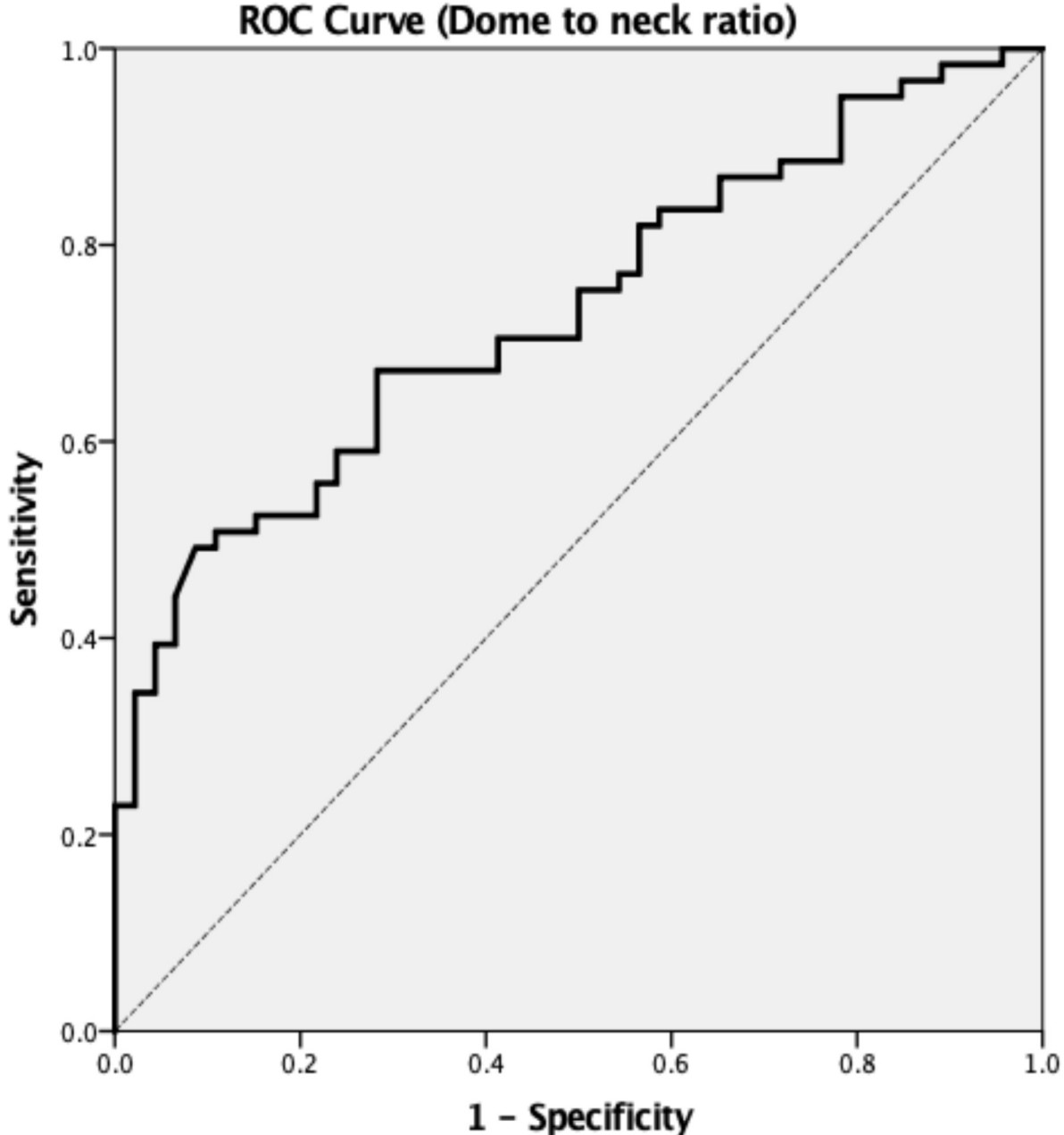

**Fig 4. The receiver operating curve of the dome to neck ratio parameter for predicting the false positive aneurysm on digital subtraction angiography.**

the true angiographic architecture frequently occur. [14] Theoretically, high-resolution equipment and immobilization during the examination are necessary to detect these small and narrow vessels; however, this approach has limited applicability in clinical settings.

### Factors associated with FPAM

In our study, FPAMs had significantly lower dome to neck ratios than true positive aneurysms (0.75±0.15 vs. 1.08±0.53, $p<0.001$). The low dome to neck ratio of saccular aneurysms is due to a wide neck morphology. These hill-shaped structures (Fig 2A) might be associated with

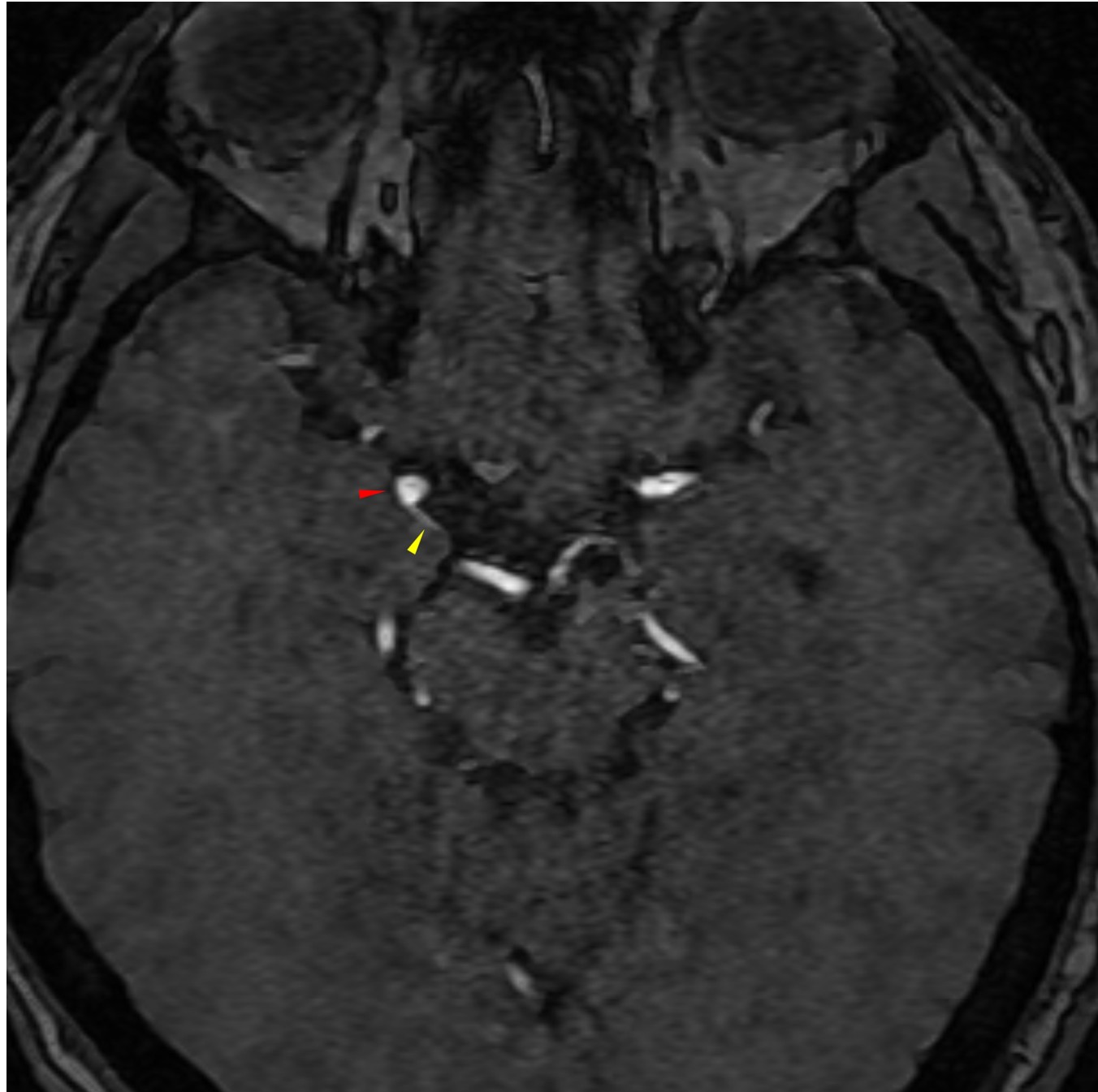

**Fig 5. Illustrative image of an infundibulum of the PCoA.** In the case of such aneurysm-like structures (red arrowhead), a junctional dilation (or infundibulum) is highly suspected as opposed to an aneurysm, because the PCoA is visible (yellow arrowhead). This lesion is not included as a suspicious aneurysm in the study. PCoA, posterior communicating artery.

tortuosity or turbulent flow of the ICA, which usually results in luminal irregularity of the vessel walls. We conducted a ROC curve analysis and determined the threshold value for the dome to neck ratio as 0.9889 (high sensitivity and specificity). Based on our results, suspicious aneurysm-like structures with a wide-neck morphology (dome to neck <0.99) may be observed through regular follow-up and do not necessitate invasive catheterized angiography.

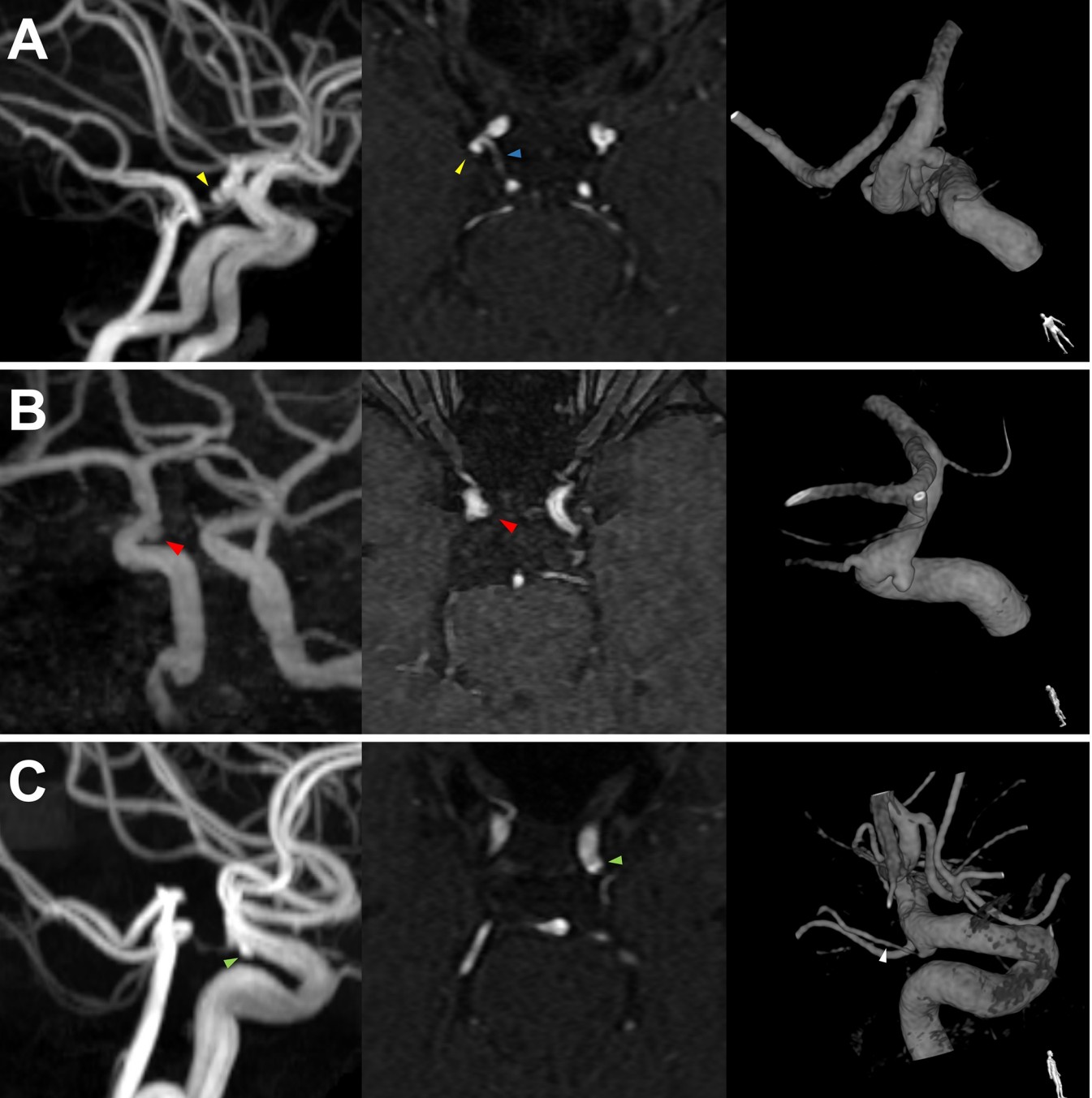

**Fig 6.** Representative cases of a side-branch aneurysm (A), side-wall aneurysm (B), and infundibulum (C). In the case of a side-branch aneurysm (yellow arrowhead), a separate normal branch (blue arrowhead) can indicate an aneurysm as opposed to a junctional dilation. In the case of a side-wall aneurysm (red arrowhead), the axial raw image shows a saccular aneurysm extruded from the main trunk. A true side-wall aneurysm was suspected in the case of (C), because the out-pouched sac (green arrowhead) was observed in the axial image without any visible branch flow. However, a narrow distal flow was observed in the catheterized angiography image (white arrowhead), and this suspicious structure was judged as a false positive on DSA.

Thirty-six of the 107 aneurysms were located on the C7 ICA, of which the majority were false positive (29/36, 80.6%, $p = 0.01$). The majority of the aneurysms located on the communicating segment of the ICA are classified as side-branch (AChA or PCoA) aneurysms, [15] and

the low flow in the side branch vessels is sometimes limited in terms of being detectable by screening tools. Considering that PCoA flow is occasionally not detected on MRA, especially when the posterior circulating flow is strong, small PCoA or AChA aneurysms may be monitored by routine follow-up using MRA.

## Clinical implications

In conclusion, suspicious aneurysm-like structures identified by MRA, located in the C7 ICA, and having a wide neck morphology (dome to neck ratio <0.99) may be monitored via regular follow-ups instead of performing invasive catheterized angiography.

As endovascular treatments and surgical techniques have improved, there are fewer barriers to treating aneurysms. Therefore, DSA is more frequently performed for angioarchitecture evaluation and treatment planning. [7] However, it is invasive and is associated with complications such as puncture site hematoma, contrast-induced kidney injury, and procedure-related infarction. Additionally, patients are required to be hospitalized, and the procedure is cost-intensive. [16] Considering these factors, small suspicious aneurysm-like structures with a wide neck morphology should be monitored due to their low rupture risk, and the possibility of administering antiplatelet agents (in case of stent-insertion) should be considered. Invasive catheterized angiography is not recommended for structures with FPAM features. [17–19]

One major discrepancy between our results and those of previous studies was in terms of the false-positive rate. As mentioned, the reported sensitivity and specificity of screening tools for detecting UIAs are both approximately 90%. [20–22] However, in our cohort, the precision rate of UIA detection was only 57%, and the false-positive rate was 43%. These differences probably occurred due to the focus on small aneurysms in this study. We included small aneurysms less than 5 mm in maximal diameter on MRA, because (1) physicians are usually in a dilemma whether to perform DSA for aneurysms in this specific size range, and (2) aneurysms larger than 5 mm are usually true positives, with most previous studies reporting over 90% sensitivity on MRA. In our hospital, most patients with small aneurysms underwent confirmative angiography because they were in the military. In the manual of the physical grading system of the Republic of Korea Army, military personnel who have suspicious intracranial aneurysms are required to undergo catheterized angiography for a definitive diagnosis. While some may criticize such liberal guidelines regarding cerebral angiography in military hospitals, they are necessary for effectively allocating personnel to the appropriate department. Furthermore, our significant experience in performing catheterized angiography for small aneurysms is informative and enabled the current study design. Individuals in the specific cohort considered in our study (healthy young male soldiers) usually have very few pathologies on their arterial wall, which enables clear MRA or DSA imaging without distortions. Our scientific conclusions are focused on geometric factors rather than patient characteristics and can be extrapolated to the general population.

## Limitations

The current study has several limitations. First, there was heterogeneity among the included aneurysms—the locations varied from the cavernous segment of the ICA to the distal cerebral arteries, and the measured parameters did not fully represent the true angioarchitecture of the aneurysms. Second, the geometrical parameters of the aneurysms were measured manually, and thus, a more objective and automated method for these measurements is needed. Third, the results of our study cannot be generalized due to the small sample size and retrospective design. A future study with a larger cohort would be more reliable and representative.

## Conclusion

Based on our results, suspicious aneurysm-like structures characterized by factors associated with FPAM, would be highly suspected of being negative on DSA. These key factors include

1. location on the communicating segment of the ICA (and not the clinoid or ophthalmic segments),

2. a wide neck morphology with a dome to neck ratio < 0.99.

## Supporting information

**S1 Data.**
(XLSX)

## Author Contributions

**Conceptualization:** Minsu Jang, Jang Hun Kim, Jin Woo Park, Han-Joo Lee, Sung Hwan Hwang.

**Data curation:** Minsu Jang, Jang Hun Kim, Sang Hoon Yoon.

**Formal analysis:** Minsu Jang, Jang Hun Kim.

**Funding acquisition:** Jang Hun Kim, Jin Woo Park.

**Investigation:** Minsu Jang, Jang Hun Kim, Joon Ho Yoon.

**Methodology:** Minsu Jang, Jang Hun Kim.

**Project administration:** Jang Hun Kim, Joon Ho Yoon.

**Resources:** Minsu Jang, Jang Hun Kim, Jin Woo Park, Han-Joo Lee.

**Software:** Jang Hun Kim, Han-Joo Lee, Junghan Seo.

**Supervision:** Jang Hun Kim, Haewon Roh, Junghan Seo, Joon Ho Yoon, Sang Hoon Yoon, Byung-Kyu Cho.

**Validation:** Jang Hun Kim, Sung Hwan Hwang.

**Visualization:** Minsu Jang, Jang Hun Kim, Jin Woo Park, Sung Hwan Hwang.

**Writing – original draft:** Jang Hun Kim, Haewon Roh.

**Writing – review & editing:** Jang Hun Kim, Byung-Kyu Cho.

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
