## [Decision Letter · Decision Letter 0]

2 Jul 2020

PONE-D-20-12943

Features of False Positive Unruptured Intracranial Aneurysms on Screening Tests of Magnetic Resonance or Computed Tomographic Angiography That Are Finally Confirmed as Negative on Digital Subtraction Angiography

PLOS ONE

Dear Dr. Jang Hun Kim,

Thank you for submitting your manuscript to PLOS ONE. After careful consideration, we feel that it has merit but does not fully meet PLOS ONE’s publication criteria as it currently stands. Therefore, we invite you to submit a revised version of the manuscript that addresses the points raised during the review process.

We look forward to receiving your revised manuscript.

Kind regards,

Massimiliano Toscano

Academic Editor

PLOS ONE

Journal Requirements:

3. Please ensure that you refer to Figure 6 in your text as, if accepted, production will need this reference to link the reader to the figure.

4. Please include a caption for figure 6.

Additional Editor Comments (if provided):

The main theme is interesting as the paper introduces a novel point of view in the scenery of unruptured intracranial aneurysms. Anyway, there are major concerns on rationale and methodology, so that substantial reviews are needed.

Reviewers' comments:

Reviewer's Responses to Questions

**Comments to the Author**

1. Is the manuscript technically sound, and do the data support the conclusions?

Reviewer #1: Yes

Reviewer #2: Partly

2. Has the statistical analysis been performed appropriately and rigorously? 

Reviewer #1: Yes

Reviewer #2: Yes

3. Have the authors made all data underlying the findings in their manuscript fully available?

Reviewer #1: Yes

Reviewer #2: Yes

4. Is the manuscript presented in an intelligible fashion and written in standard English?

Reviewer #1: No

Reviewer #2: Yes

5. Review Comments to the Author

Reviewer #1: In this interesting article by Jang and colleagues the well common dilemma of suspicious aneurysm-like structures that eventually are not identified at cerebral angiography is discussed. The Authors reviewed a series of 121 suspicious-for-aneurysms structures, 52 of whom were indicated a posteriori on angiography as being false positives. Univariate and multivariate analysis identified some factors that may be linked to false positives on intracranial angiographies.

The main theme is interesting; introduction and material and methods sections are adequate and sufficiently concise; the number of patients analyzed is consistent, although suffering from a very strong selection bias(military cohort, see in the Discussion section). Looking at this aspect, I particularly appreciated the choice to concentrate the analysis on purely aneurysmal pathologies, excluding aneurysms in association with other vascular diseases (i.e. arteriovenous malformations, moyamoya disease…).

Some other concerns that, to me, should be addressed before considering the work for possible publication:

- do the Authors choose to study aneurysms < 5mm because the rate of false positives for such dimensions is higher? How do the Authors reputate that this choice may have influenced the results of the study? As this aspect has been raised also from Reviewer 3 in a previous revision of the same manuscript, I think a specific Discussion around this point should be included in the Discussion section;

- were readers of the radiological images blinded each other to the results of the other colleague? Please, specify;

- in the present form, the article appears comprehensible concerning its grammar/syntax form and its general meaning, considering the main suggestions of Reviewer 1. Nevertheless, it would beneficiate from a further strengthening of its grammar form by a mother-tongue speaker.

- In the Limitations section, what does it mean that “the interpretation of the result need cautions”? Which cautions? Please specify and explain this fundamental point, as one of the major limitation of this work, that may limit its applicability to general population, resides in the highly selected population.

Reviewer #2: Thank you for the opportunity to review this interesting manuscript which introduces a novel point of view in the scenery of unruptured intracranial aneurysms. Unfortunately, the study suffers from concerns on rationale and methodology that limit the enthusiasm.

In fact, a major concern on study rationale remains. Indeed, authors aimed to establish the ratio between false positive aneurysms initially detected at computed tomography angiography (CTA) and magnetic resonance angiography (MRA) screening and true positive aneurysms confirmed at digital subtraction angiography (DSA). I understand the peculiar clinical setting according to Korean military hospital. Authors claimed that in this setting it is mandatory to exclude the presence of aneurysms in asymptomatic subjects. On the contrary, clinical efforts are directed toward detecting false negative aneurysms which are clinically more relevant with the view to preventing subarachnoid hemorrhage.

The most prevalent false aneurysm-like structures are in the ICA, and more specifically in the C7 ICA, which is the less risky location for aneurysm rupture, even more because aneurysms are of small size (<5 mm). As already underlined by previous reviewers, indication for DSA in this population seems to be debatable and, even if mandatory for militaries, the generalizability of results is limited. Moreover, it is no clear the reason why subjects received cerebral angiography at baseline.

Another concern regards the use of CTA ad MRA indifferently as a screening tool. The two methods present slightly different sensitivity, and this could affect results of the study. Moreover, since the vast majority (~85%) of subject underwent MRA, it would be better to exclude from the analysis patients who underwent CTA.

The use of term “false-positive aneurysms structures in screening tools” is confusing. Why did authors not consider the use of term “suspicious aneurysms” or “probable aneurysm”?

The title could be shortened to “Features of false positive unruptured intracranial aneurysms on screening tests of magnetic resonance or computed tomographic angiography” avoiding the final part. The use of “false positive” implies the presence of a control DSA as it represents the gold standard technique. “Tests” should be replaced with “tools” or “techniques”.

6. PLOS authors have the option to publish the peer review history of their article (what does this mean?). If published, this will include your full peer review and any attached files.

Reviewer #1: **Yes: **Francesco Acerbi

Reviewer #2: No

---

## [Author Response · Author response to Decision Letter 0]

14 Jul 2020

PONE-D-20-12943

Features of False Positive Unruptured Intracranial Aneurysms on Screening Tests of Magnetic Resonance or Computed Tomographic Angiography That Are Finally Confirmed as Negative on Digital Subtraction Angiography

PLOS ONE

Additional Editor Comments (if provided):

The main theme is interesting as the paper introduces a novel point of view in the scenery of unruptured intracranial aneurysms. Anyway, there are major concerns on rationale and methodology, so that substantial reviews are needed.

On behalf of my co-authors, I would like to extend our gratitude and appreciation to you and the reviewers for your thorough evaluation of our manuscript and for providing insightful and constructive comments and suggestions which have significantly helped us to improve our manuscript. We have revised the manuscript accordingly and hope that the revised version of the manuscript meets your standards and can be reconsidered for publication in PLOS One.

We have also provided point-by-point responses to your comments and questions, and we hope that our replies address all your concerns. 

Should you have any further questions or concerns, please contact us. Thank you for your consideration. We look forwards to hearing from you.

Sincerely, 

Jang Hun Kim, M.D.

 

Reviewer #1: In this interesting article by Jang and colleagues the well common dilemma of suspicious aneurysm-like structures that eventually are not identified at cerebral angiography is discussed. The Authors reviewed a series of 121 suspicious-for-aneurysms structures, 52 of whom were indicated posteriori on angiography as being false positives. Univariate and multivariate analysis identified some factors that may be linked to false positives on intracranial angiographies.

The main theme is interesting; introduction and material and methods sections are adequate and sufficiently concise; the number of patients analyzed is consistent, although suffering from a very strong selection bias(military cohort, see in the Discussion section). Looking at this aspect, I particularly appreciated the choice to concentrate the analysis on purely aneurysmal pathologies, excluding aneurysms in association with other vascular diseases (i.e. arteriovenous malformations, moyamoya disease…).

 Thank you for your comments and for the evaluation of our manuscript. Once again, I would like to extend our gratitude and appreciation to you for providing insightful and constructive comments and suggestions which have significantly helped us to improve our manuscript. 

Some other concerns that, to me, should be addressed before considering the work for possible publication:

- do the Authors choose to study aneurysms < 5mm because the rate of false positives for such dimensions is higher? How do the Authors reputate that this choice may have influenced the results of the study? As this aspect has been raised also from Reviewer 3 in a previous revision of the same manuscript, I think a specific Discussion around this point should be included in the Discussion section;

We generally agree with your comments. We chose small aneurysms (<5 mm in maximal size) because (1) physicians usually are in a dilemma whether or not to perform DSA for aneurysms in this size range, and (2) aneurysms larger than 5 mm usually are “true positives”, with most previous studies reporting over 90% sensitivity on MRA.

Actually, there is no definitive guideline about the selective indication of performing DSA. Instead, DSA is usually performed for planning, especially when the aneurysm is to be treated either by coil embolization or surgical clipping. Although the specific size may be debatable, aneurysms usually larger than 5 mm are indicated for treatment. These are also the ones that are usually confirmed as “true” on DSA. In this perspective, DSA is performed not for confirmative diagnosis but for planning surgery or embolization. When physicians encounter suspicious aneurysms larger than 5 mm, they do not hesitate to perform DSA and plan treatment. In this study, we focused on the smaller aneurysms which can put physicians in a dilemma regarding whether DSA is needed for confirmative diagnosis. 

Also, as mentioned in the Discussion section, there might be discrepancies between the false positive rates of previous studies and that of our current study, because the included aneurysms were different. However, we did not focus on the false positive rate in this study; instead, we tried to identify the risk factors or predictors indicative of negative aneurysms on DSA. Therefore, we conducted multivariate logistic regression analysis, which revealed location on C7 ICA and a lower dome to neck ratio as predictors of DSA-negative aneurysms. A more extensive cohort may result in differences in false positive rates; however, the ‘risk factors’ might be similar because of the characteristics of multivariate logistic regression analysis. 

With regard to your concerns, we have added the following sentences in Discussion section: We included small aneurysms less than 5 mm in maximal diameter on MRA, because (1) physicians are usually in a dilemma whether to perform DSA for aneurysms in this specific size range, and (2) aneurysms larger than 5 mm are usually true positives, with most previous studies reporting over 90% sensitivity on MRA.

- were readers of the radiological images blinded each other to the results of the other colleague? Please, specify;

Thank you for your advice. MRI findings were evaluated separately by one radiologist and one neurosurgeon, with mediation by a senior neurosurgeon if required. Thereafter, DSA findings were assessed separately by two neurosurgeons blinded to each other’s observations. There were no discrepancies regarding the DSA assessments. According to your concerns, we have modified the relevant text in the manuscript as follows: Judgements regarding the radiologic images were separately made by two physicians (JH Kim & MS Jang) who were blinded to each other’s observations. No discrepancies were noted.

- in the present form, the article appears comprehensible concerning its grammar/syntax form and its general meaning, considering the main suggestions of Reviewer 1. Nevertheless, it would beneficiate from a further strengthening of its grammar form by a mother-tongue speaker.

We generally agree with your comments. Accordingly, the manuscript was rechecked by a native English speaker. 

- In the Limitations section, what does it mean that “the interpretation of the result need cautions”? Which cautions? Please specify and explain this fundamental point, as one of the major limitations of this work, that may limit its applicability to general population, resides in the highly selected population.

We really appreciate your advice. Actually, we do not believe that the enrolled population limits the scientific conclusion of the study, because it was a very homogenous, young, and healthy male cohort, and such individuals usually have very few pathologies on the arterial wall; thus, there are almost no distortions on MRA. We focused on the geometric factors of angiograms rather than on patient demographic factors such as age, sex, and medical history. Therefore, our results can be extrapolated to the general population, because the conclusive factors are less affected by population or cohort differences. 

However, as the previous Reviewer 2 had doubts about our study cohort and had raised some concerns, we had added a sentence in limitation section. Now, we have deleted the mentioned limitation from the text. Furthermore, we have added a paragraph in the Discussion section on this aspect: Individuals in the specific cohort considered in our study (healthy young male soldiers) usually have very few pathologies on their arterial wall, which enables clear MRA and DSA imaging without distortions. Our scientific conclusions are focused on geometric factors rather than patient characteristics and can be extrapolated to the general population.

 

Reviewer #2: Thank you for the opportunity to review this interesting manuscript which introduces a novel point of view in the scenery of unruptured intracranial aneurysms. Unfortunately, the study suffers from concerns on rationale and methodology that limit the enthusiasm.

In fact, a major concern on study rationale remains. Indeed, authors aimed to establish the ratio between false positive aneurysms initially detected at computed tomography angiography (CTA) and magnetic resonance angiography (MRA) screening and true positive aneurysms confirmed at digital subtraction angiography (DSA). I understand the peculiar clinical setting according to Korean military hospital. Authors claimed that in this setting it is mandatory to exclude the presence of aneurysms in asymptomatic subjects. On the contrary, clinical efforts are directed toward detecting false negative aneurysms which are clinically more relevant with the view to preventing subarachnoid hemorrhage.

First, we appreciate your comments and the thorough evaluation of our manuscript. We would also like to extend our gratitude and appreciation to you for providing insightful and constructive comments which have significantly helped us to improve our manuscript. Regarding your concerns, our study is unique with regard to the included population of soldiers, and we thank you for understanding the peculiar setting. Actually, we did not think the enrolled population limits the scientific conclusions of the study, because it consisted of very homogenous, young, and healthy male soldiers. Such individuals usually have very few pathologies on the arterial wall, and thus, the MRA images usually have no distortions. We focused on the geometric factors of angiograms rather than on patient characteristics, such as, age, sex, and medical history. Our results can be adapted to the general populations, because the factors we found are less affected by population or cohort differences.

Indeed, the current study does not aim to establish the ratio of false positive rate but focused on the risk factors or predictors of false positive aneurysms. We are aware of the liberal indications of DSA have also encountered a lot of false positive cases on DSA. Here, we did not want to discuss the details of these experiences but tried to figure out the risk factors for DSA-negative aneurysms. Maybe the mention of the unique setting led to the conclusions. Therefore, the discrepancy between the false positive rates of previous studies and that of the current one is to be expected, because the aneurysms included in our study were much smaller than those in the previous one. We discuss this in the Discussion section as follows: One major discrepancy between our results and those of previous studies was in terms of the false-positive rate. As mentioned, the reported sensitivity and specificity of screening tools for detecting UIAs are both approximately 90%.21-23 However, in our cohort, the precision rate of UIA detection was only 57%, and the false-positive rate was 43%. These differences probably occurred due to the inclusion of small aneurysms.

The most prevalent false aneurysm-like structures are in the ICA, and more specifically in the C7 ICA, which is the less risky location for aneurysm rupture, even more because aneurysms are of small size (<5 mm). As already underlined by previous reviewers, indication for DSA in this population seems to be debatable and, even if mandatory for militaries, the generalizability of results is limited. Moreover, it is no clear the reason why subjects received cerebral angiography at baseline.

We largely agree with your comments. Actually, there is no definitive guideline about selective indications for DSA. Instead, it is usually used at the planning stage, especially when the aneurysm can be treated either by coil embolization or surgical clipping. Although the specific size for treatment indication may be debatable, usually aneurysms larger than 5 mm are relatively indicated for treatment. When physicians encounter suspicious aneurysms larger than 5 mm, they do not hesitate to perform DSA. In this situation, DSA is performed not for confirmative diagnosis but for planning surgery or embolization. We focused on the small aneurysms which can put physicians in a dilemma regarding whether or not DSA is needed. Some may criticize the liberal guideline of our institution; however, the scientific results can be a good reference from the general clinical perspective. The results of the current study might be helpful in the out-patient clinic setting where physicians can encounter a suspicious lesion and strengthen the rationale on un-necessary DSA procedures. Further, our peculiar military setting may be similar to that of patients having a strong family history or insurance problems. These patients may want to undergo DSA in spite of the low clinical importance. Moreover, the identification of suspicious lesions can be a reference for future follow-ups. 

Another concern regards the use of CTA ad MRA indifferently as a screening tool. The two methods present slightly different sensitivity, and this could affect results of the study. Moreover, since the vast majority (~85%) of subject underwent MRA, it would be better to exclude from the analysis patients who underwent CTA.

We generally agree with your comments. Accordingly, we excluded the patients who underwent CTA and reanalyzed the data; please refer to the modified tables and results in the revised manuscript. The result of the reanalysis was different in that height was not verified as a significant factor in the multivariate analysis due to the small sample size. 

The use of term “false-positive aneurysms structures in screening tools” is confusing. Why did authors not consider the use of term “suspicious aneurysms” or “probable aneurysm”?

The title could be shortened to “Features of false positive unruptured intracranial aneurysms on screening tests of magnetic resonance or computed tomographic angiography” avoiding the final part. The use of “false positive” implies the presence of a control DSA as it represents the gold standard technique. “Tests” should be replaced with “tools” or “techniques”.

We generally agree with your comments. Before DSA was performed, any suspicious aneurysm-like structure was termed as a “suspicious aneurysm.” After confirmative DSA, each structure was classified as a “false” or “true” positive aneurysm depending on the results of the 3D angiogram. 

Furthermore, we have modified the title of the article to “Features of “False Positive” Unruptured Intracranial Aneurysms on Screening Magnetic Resonance Angiography.”

---

## [Decision Letter · Decision Letter 1]

20 Aug 2020

Features of “False Positive” Unruptured Intracranial Aneurysms on Screening Magnetic Resonance Angiography

PONE-D-20-12943R1

Dear Dr. Kim,

We’re pleased to inform you that your manuscript has been judged scientifically suitable for publication and will be formally accepted for publication once it meets all outstanding technical requirements.

Kind regards,

Domokos Máthé

Academic Editor

PLOS ONE

Additional Editor Comments (optional):

Reviewers' comments:

Reviewer's Responses to Questions

**Comments to the Author**

1. If the authors have adequately addressed your comments raised in a previous round of review and you feel that this manuscript is now acceptable for publication, you may indicate that here to bypass the “Comments to the Author” section, enter your conflict of interest statement in the “Confidential to Editor” section, and submit your "Accept" recommendation.

Reviewer #1: All comments have been addressed

Reviewer #3: All comments have been addressed

2. Is the manuscript technically sound, and do the data support the conclusions?

Reviewer #1: Yes

Reviewer #3: Yes

3. Has the statistical analysis been performed appropriately and rigorously? 

Reviewer #1: Yes

Reviewer #3: Yes

4. Have the authors made all data underlying the findings in their manuscript fully available?

Reviewer #1: Yes

Reviewer #3: Yes

5. Is the manuscript presented in an intelligible fashion and written in standard English?

Reviewer #1: Yes

Reviewer #3: Yes

6. Review Comments to the Author

Reviewer #1: (No Response)

Reviewer #3: Clinically interesting initial exploration of a relevant topic. The points are well written, results provide evidence to possibly reduce the usage of invasive diagnostic methods. The revised manuscript sufficiently addresses the initial concerns. The scientific message is enhanced further with this revision. I propose its acceptance.

7. PLOS authors have the option to publish the peer review history of their article (what does this mean?). If published, this will include your full peer review and any attached files.

Reviewer #1: **Yes: **Francesco Acerbi, MD PhD

Reviewer #3: No

---

## [Editor Report · Acceptance letter]

24 Aug 2020

PONE-D-20-12943R1 

Features of “False Positive” Unruptured Intracranial Aneurysms on Screening Magnetic Resonance Angiography 

Dear Dr. Kim:

I'm pleased to inform you that your manuscript has been deemed suitable for publication in PLOS ONE. Congratulations! Your manuscript is now with our production department. 

Kind regards, 

on behalf of

Dr. Domokos Máthé 

Academic Editor

PLOS ONE